# Change in Stress Among Individuals with Chronic Health Conditions and Low Health Literacy Using a Tailored Communication App Promoting Self-Management

**DOI:** 10.3390/bs15060783

**Published:** 2025-06-06

**Authors:** Raymond L. Ownby, Joshua Caballero

**Affiliations:** 1Department of Psychiatry and Behavioral Medicine, Nova Southeastern University, Fort Lauderdale, FL 33328, USA; 2Department of Clinical and Administrative Pharmacy, University of Georgia, Athens, GA 30602, USA; joshua.caballero@uga.edu

**Keywords:** tailored information, health literacy, chronic disease self-management

## Abstract

Chronic disease self-management (CDSM) is critical for improving health outcomes, particularly for individuals with low health literacy who face barriers in accessing and understanding medical information. This study examined the role of tailored digital health interventions in reducing stress and improving quality of life among individuals with chronic conditions. A multisite randomized controlled trial including 309 participants at baseline aged 40 to 90 years was conducted using a mobile app that provided self-management information at different literacy levels. Latent growth curve modeling (LGM) computed in Mplus was used to assess changes in stress over time and its impact on quality of life. The results indicated that successful tailoring of health information was significantly associated with reductions in perceived stress, which, in turn, predicted improvements in quality of life. These findings suggest that personalized digital interventions may enhance engagement with self-management strategies, alleviate psychological distress, and contribute to better overall well-being. This study highlights the importance of tailoring health content to users’ literacy levels and informational needs, underscoring the potential of mobile health solutions for improving CDSM. Future research should explore additional mechanisms underlying these effects and assess the long-term sustainability of digital interventions in diverse populations. These results provide valuable insights into optimizing mobile health applications to support individuals with chronic conditions.

## 1. Introduction

Chronic health conditions significantly impact individuals’ daily lives and may place a burden on healthcare systems. According to global estimates, chronic diseases, such as diabetes, cardiovascular diseases, mental health, and respiratory conditions, account for a high proportion of disability and mortality worldwide. Many older adults have multiple chronic conditions, known as multimorbidity, which can increase the number of medications taken daily (e.g., polypharmacy). Data suggest an association between increasing age and clinical complexity ([31]). Increasing polypharmacy may negatively impact adherence and health outcomes ([12]). For example, studies have shown that polypharmacy may increase the risk of falls and hospitalizations ([15]). These conditions are more common among individuals with lower socioeconomic status and minority groups, who are disproportionately affected due to systemic barriers in accessing and utilizing healthcare resources ([3]; [38]; [40]). Data also suggest that individuals with chronic disease and lower socioeconomic status may experience increased psychological distress (e.g., loneliness, isolation, lack of support) and lower quality of life ([47]). These challenges highlight the importance of effective strategies to improve chronic disease self-management (CDSM), particularly for individuals with limited health literacy.

Managing chronic diseases may involve complex medication regimens and lifestyle adjustments, which can be daunting for patients, especially those with low health literacy. For example, approximately 70% of adults (40–79 years of age) in the United States and Canada take at least one prescription medication, and about 20% use at least five prescription medications ([19]). Limited health literacy is a documented barrier to understanding health information, following medical instruction, and engaging in effective self-management behaviors. A recent study reported that individuals with lower health literacy had an increased number of chronic conditions ([49]). For patients with chronic conditions, this can lead to frustration, lowered self-esteem and confidence managing their health, and increased stress. While traditional education approaches, such as pamphlets or in-person training, have shown some effectiveness, their reach can be limited by factors including cost, geographic availability, and lack of trained professionals. The results are a call to action by researchers who advocate the importance of not just focusing on individuals but also understanding the complexity of the interaction between patients and healthcare systems, such as accessibility, communication, and the difficulty level of information ([46]).

CDSM programs have reported favorable outcomes in equipping individuals with the knowledge and skills needed to manage their conditions ([22]). A review of the literature suggests that self-management programs may offer better outcomes than routine care and may improve self-efficacy and quality of life and reduce depressive symptoms ([22]). However, in-person programs are not widely accessible, creating a need for alternative delivery methods. eHealth applications represent a promising solution, offering scalable, personalized, and interactive tools for disease management ([23]). By tailoring content to individuals’ needs and health literacy levels, these apps can bridge the gap between the availability of CDSM programs and the needs of patients.

Having positive relationships with relatives, friends, peers, or healthcare providers may provide emotional support to improve CDSM ([41]; [44]). For example, studies in adults 40 years of age or older have shown that CDSM peer instructor programs lead to improvements in health distress and health behaviors ([29]). Additionally, a meta-analysis suggested small–moderate improvements when small English-speaking groups are used to deliver CDSM ([6]). However, friends, family, and healthcare providers may be ill equipped or lack proper training to support CDSM. For example, over 40% of friends and family report a lack of education and information to properly support individuals with chronic diseases ([26]). Studies assessing the provider–patient relationship appear to show that healthcare providers remain in an authoritative position, limiting the ability of patients to express concerns ([14]). Such behaviors that show a lack of empathy by healthcare practitioners may result in decreased communication with patients, which may lead to poorer health outcomes, such as lower medication adherence ([16]). Therefore, while CDSM programs appear to show positive outcomes, real-life scenarios suggest otherwise. There is, thus, a need to explore other interventions to empower patients to manage their chronic health conditions.

Different types of stress can negatively impact health behaviors and outcomes. These stresses can include acute (e.g., short-term infection, accident), chronic (long-term illness), social/environmental (e.g., discrimination, poverty), psychological (e.g., depression, anxiety), and physical (e.g., pain, cancer, disability). These types of stress can decrease motivation to adhere to medications and treatment programs and develop negative coping skills, such as drinking and inappropriate substance use. These factors may all be intertwined and create a cumulative biological burden known as allostatic load. For example, acute or chronic pain can induce stress that may increase depressive or anxiety symptoms and negatively affect sleep, which may result in an elevated risk in nonadherence to other chronic illnesses, such as diabetes or hypertension. Over time, the prolonged and multiple stressors (i.e., allostatic load) may worsen health outcomes. Data suggest that engaging in self-management programs can reduce stress, improve mental health, and increase quality of life ([4]). However, other studies suggest eHealth interventions may not reduce stress ([23]). For example, a study assessing a digital multi-media health promotion reported that the digital promotion was not more effective in reducing stress compared to printed materials ([10]). The authors discussed that this may be due to several factors, including a lower than anticipated usage of the web-based platform.

It is thus imperative that such programs are delivered at the appropriate level to maximize their effectiveness. For example, information should be tailored to the appropriate health literacy level but also targeted to a patient’s specific need ([35], [36]) and culturally adapted to improve usability and acceptability. Digital applications apply these constructs to develop content that may not only manage stress but promote disease self-management. Therefore, providing patients with tailored information through a digital platform has the potential to improve their understanding of disease management and enhance their confidence in implementing self-care strategies.

In this study, we explored the relationship between perceived personalization of app-delivered information, stress reduction, and self-management outcomes, drawing on data from a previous study. In the main outcome analyses in this study, we did not observe differences in outcomes between participants in experimental and control groups, even though all groups showed improvements on most of the main outcome variables ([36]). This left open the question of whether providing individually tailored information had an impact on study outcomes. In the analyses reported here, we assess the relations between successfully tailoring information and change in stress and between change in stress and quality of life at study end in order to address this question.

## 2. Materials and Methods

### 2.1. App Development

A mobile application was developed to support individuals with chronic health conditions in managing their health and has been described in detail elsewhere ([36]). Users were involved in the app development, which was informed by the results of a qualitative study ([24]) and multiple rounds of usability testing. The app included three distinct versions, each providing identical content at different reading levels, 8th grade, 6th grade, and 3rd grade, with the latter including audio narration. The content was designed to address key aspects of CDSM, such as medication adherence, stress management, symptom monitoring, and communication with healthcare providers. A multidisciplinary team, including experts in medicine, pharmacy, public health, nursing, psychology, and education, collaborated to ensure the content was evidence-based and culturally appropriate. Iterative usability testing was conducted with target users to refine both the content and the user interface, ensuring accessibility for individuals with low health literacy.

Each module of the app addressed one of the aspects. For example, the stress module began with a series of questions about the participant’s experience of stress (e.g., feeling that he or she had too many unexpected challenges, feeling out of control, feeling overwhelmed). The module was programmed to then provide general information about stress (e.g., the role of cortisol in stress) and its potential effects on health. The app was then programmed to provide individually tailored feedback about the participant’s previous responses to questions about stress, first repeating what was said (“You said you often feel overwhelmed”) and empathic acknowledgement (“That’s too bad, that must be really hard for you”) and then specific suggestions about how to cope with the feeling (“Maybe you could try stepping back and taking a few deep breaths.” Additional details on the development of the intervention are provided in previous publications ([37]). The content of the intervention was distributed over three sessions. The first day included an introduction to the app, its purpose, and its functioning and modules on adherence and stress. The second day focused on sleep, mood, pain, and memory, while the third day focused on fatigue, shortness of breath, and anger.

### 2.2. Study Design

The multisite randomized controlled trial from which the data presented here are drawn evaluated the effectiveness of the app in improving key outcomes for individuals aged 40 years and older with low health literacy and at least one chronic health condition ([35], [36]). The participants were randomized to one of the three app versions. Intervention sessions were conducted over a two- to three-week period, with each participant completing three sessions. The sessions were designed to engage the participants with the app’s interactive modules using tablet devices, which were selected for their readability and ease of navigation.

### 2.3. Participant Recruitment and Screening

The participants were recruited through community outreach efforts, healthcare clinics, and prior research databases. The eligibility criteria included being 40 years of age or older, having a chronic health condition requiring medical treatment, and scoring below a validated health literacy threshold on a short form of the Rapid Estimate of Adult Literacy in Medicine. Since the focus of the intervention was on individuals with chronic health conditions, only individuals 40 years of age and older were included because of the low prevalence of chronic health conditions in younger individuals. Screening involved assessments of health literacy, reading comprehension, and sensory acuity to confirm the participants’ ability to interact with the app’s features. No exclusion criteria were applied, except if the investigators at each site (both experienced clinicians) judged that the person was not able to provide informed consent for their participation. Chronic conditions were defined as a diagnosis and current treatment for one of the items in the Functional Comorbidity Index ([18]) because the Index was defined as including conditions related to physical function. It was expanded to include a number of conditions requiring treatment that are common in US Medicare statistics, such as hypertension and dyslipidemia ([7]).

### 2.4. Outcome Measures

Primary outcomes included patient activation measured using the Patient Activation Measure ([21]), chronic disease self-efficacy assessed with the Chronic Disease Self-Efficacy Scale ([28]), health-related quality of life evaluated using the Short Form-36 General Health subscale ([48]), and self-report of medication adherence ([17]). These measures were selected based on their relevance to chronic disease management and their strong psychometric properties. Primary study outcomes have been completely reported elsewhere ([36]). Briefly, we found that activation, quality of life, and self-efficacy but not adherence improved over the course of the study for all groups. No between-group differences were observed, leading to a question of what might have been responsible for the observed changes in outcomes over time ([34]). As originally hypothesized, the extent to which information was successfully tailored to the participants’ interests and needs was believed to be a key factor in how information influenced the study outcomes, based on the Elaboration Likelihood Model of information processing ([39]).

Prior to study initiation, a search was completed for a measure that could assess the extent to which the participants believed the information presented to them was relevant, helpful, and actionable. A scale was developed to assess dimensions of the relevance of the information to the participants based on a literature review on relevance ([11]) while taking into account the Elaboration Likelihood Model ([39]). We thus aimed to assess the extent to which the information presented was perceived as related to the person, was appropriate to their individual situation, and was usable and whether they intended to act on it. The measure comprised eight items rated on a five-point Likert-type scale. While we were not able to complete additional assessments of the scale’s validity, we strove to ensure at least good face validity. Items include, for example, “The information in the modules was about people like me,” “The information in the modules was useful to someone like me,” and “The information in the modules would work for me.” Previously presented analyses showed that the scale had good internal reliability (Cronbach’s alpha = 0.95).

In addition to the primary outcome measures, we explored a number of secondary outcomes related to the app’s content. As part of the original study design ([35]), multiple secondary outcome measures were used to assess relevant variables over the course of the study. Also, as part of the study design, exploration of potential mediators and moderators of study outcomes was planned.

Here we report on the participants’ stress levels over the course of the study, as assessed by the Perceived Stress Scale ([8]). The 10-item Perceived Stress Scale (PSS-10) is a widely used psychological instrument for assessing the degree to which individuals perceive their lives as stressful over the past month. It measures feelings of unpredictability, uncontrollability, and overload using a series of statements rated on a 5-point Likert scale. We also focus on the relation of changes in stress to overall health-related quality of life, as assessed by the General Health subscale of the Medical Outcomes Study, Short Form 36 ([48]). The SF-36 is a 36-item questionnaire designed to assess overall health-related quality of life across eight domains, including physical functioning, bodily pain, vitality, and mental health. It provides a comprehensive profile of functional health and well-being from the patient’s perspective. The purpose of these analyses was to assess whether changes in stress over time were related to the successful tailoring of information provided to the participants and, in turn, related to changes in health-related quality of life, as assessed by the General Health subscale of the MOS SF-36 ([48]).

### 2.5. Statistical Analysis

Descriptive statistics were calculated using the Statistical Package for the Social Sciences, version 28 (Armonk, NY, USA: IBM Corporation). In order to test the hypothesis that change in stress was related to the hypothesized mechanism by which the app works, effective personalization of information, a latent growth curve model (LGM) was created in the Mplus statistical software, version 8.8 ([33]) in several model building stages. In the first stage, change in stress over the course of the study was evaluated in the LGM by, first, only assessing change in stress over time. We encountered computational difficulties with this model, requiring additional exploration. Examination of detailed estimates of model parameters showed incorrect values for some, including a negative covariance for an estimated parameter. Consistent with others’ recommendations for dealing with this kind of estimation issue ([50]), we fixed this covariance to a small positive value for the remaining stages of the study. Because of the likelihood of nonlinear growth over the course of the study, inferred from inspection of the plot of the measure’s actual values at each time point, we fixed the first slope coefficient to zero and the second to one while allowing the third coefficient to be freely estimated.

In the second stage, the effects of possible confounding variables on observed change were evaluated by regressing both the intercept and slope of the LGM on age, gender, race, and level of health literacy. Only age was significantly related to the participants’ baseline levels of stress (the model intercept) and was retained in subsequent analyses. We note that only age was significantly related to the main study outcomes; we did not observe significant gender- or race-related differences ([36]).

In the third stage, the impact of the participants’ reports of the extent to which information had been successfully tailored to their interests and needs was evaluated by regressing the model’s change over time (slope factor) on the Success in Tailoring Scale (SIT) described above. In the final stage, the impact of the change in stress on quality of life (SF-36 General Health scale) at study conclusion was assessed by regressing the scale score on the change in stress over time (slope factor). To further assess the possibility that successful tailoring of information was related to change in stress and that change in stress was related to quality of life, the indirect effect of tailoring on quality of life was calculated.

Missing data were handled using full information maximum likelihood estimation. This strategy has repeatedly been shown to be robust ([13]).

### 2.6. Ethics Approval

This study was conducted in accordance with the Declaration of Helsinki and approved by the Institutional Review Boards (or Ethics Committee) of Nova Southeastern University (protocol code 02261021 and 8 August 2018) and Emory University (protocol code MODCR001-IRB00087112 and 1 June 2020). All participants provided oral consent prior to screening procedures and written informed consent before study participation.

## 3. Results

Descriptive statistics for the baseline study sample are presented in Table 1.

In stage one of the model building process, change over time in stress was determined by using a growth curve model that only included the observed values for the PSS at baseline and two follow-up assessments. This model fit the data moderately well, with a nonsignificant chi-square value (χ^2^ [1] = 2.82, *p* = 0.09), a confirmatory fit index of 0.99, and a standardized root mean square residual of 0.07.

In the second stage, potential confounding variables were included in the model, including age, gender, race, and level of health literacy, with the model intercept and slope regressed on all of these. Of these variables, only age was significantly related to the model intercept (−0.07, standard error 0.03, z = −2.83, *p* = 0.005), suggesting that the baseline level of stress was inversely related to age. It was included in subsequent analyses. We note that we did not observe race- or gender-related differences in main study outcomes.

In the final stage of building, the change in time of the PSS (slope) was regressed on the extent to which the participants reported that the information in the intervention was relevant to them, and quality of life at study end (SF-36 General Health subscale) was regressed on the change in stress (slope; see Figure 1) to evaluate the extent to which successfully tailoring information to the participants was related to change and stress and, in turn, whether change in stress was related to improved quality of life. As an additional test, the specific indirect effect of successful tailoring (SIT score) to quality of life (SF-36) by way of change in stress (slope) was calculated.

This final model fit the data well, with a nonsignificant chi square value (χ^2^ [9] = 11.40, *p* = 0.25), a root mean square error of approximation less than 0.05 (0.025, 95% CI [0.00–0.07], and a confirmatory fit index greater than 0.95 (0.98). The model parameters used to evaluate the study hypothesis showed that change over time (slope factor) was significantly related to successful tailoring of information to the participants (Figure 1). Quality of life at study end was significantly related to change in stress. The indirect effect of success in tailoring information on quality of life was also significant.

## 4. Discussion

The purpose of these analyses was to explore the extent to which successfully tailoring information to the study participants’ needs and interests was related to change in their level of stress, notoriously in individuals with low health literacy and limited economic resources who are coping with chronic health conditions ([25]; [30]). Evaluation of the main study outcomes showed improvements in activation, self-efficacy, and quality of life, but no differences were observed between the control and experimental groups ([34]). This failure to find between-group differences left open the question of whether tailoring information, the hypothesized active ingredient in the intervention, might be related to observed changes in outcomes or whether simply participating in the study may have improved observed outcomes, as has been observed in other trials ([9]). The purpose of these analyses was to assess whether change in a secondary outcome (self-report of stress) was seen, whether change was related to the extent that information was successfully tailored, and then, in turn, whether changes in stress were related to health-related quality of life at study outcome. The results of the analyses presented here suggest that the intervention effects related to tailoring may have had a positive impact on the participants’ stress and, in turn, on their quality of life.

These findings are consistent with other studies of the effects of tailoring information on health-related outcomes ([1]; [20]; [42]). These results suggest that successful tailoring of self-management information within a mobile app is associated with reduced stress levels, which, in turn, contribute to improved quality of life among individuals with chronic conditions. While prior research has established the benefits of self-management programs ([22]), this study extends the literature by demonstrating the specific role of a digital intervention providing tailored information in stress reduction. The significant indirect effect of tailoring on quality of life via stress reduction supports the idea that personalized health information may help mitigate the psychological burdens associated with chronic disease. These findings align with existing studies showing that interventions adapted to individuals’ specific needs and literacy levels can enhance engagement and self-efficacy ([42]).

This study highlights that providing information at the appropriate health literacy level may be a step towards reducing stress. However, digital health apps to treat chronic conditions have focused on directly providing content to reduce stress. For example, focus groups in the development of a diabetes app recommended stress management features ([5]), while others have tested app-assisted biofeedback to modulate stress ([32]). While these interventions show varying levels of success, it is important to keep in mind that the ability for a mobile app to increase patient activation while tailoring the information to a patient’s need at the appropriate health literacy level inherently provides a reduction in stress. Therefore, future studies need to be able to determine if these additional stress management tools truly reduce stress compared to the natural ability for the digital app to reduce stress by empowering the patient with information to manage their chronic illness.

The limitations of the original study and these analyses should be acknowledged. The study design did not include a traditional control group that received only general, non-tailored health information, making it difficult to fully isolate the effects of personalization from general engagement with the app. Additionally, while stress reduction was identified as a key mechanism in quality of life, other factors such as increased patient activation or changes in self-efficacy may also have contributed to observed improvements. Our sample was selected to have low levels of health literacy but also had limited economic resources and was largely Black. This may limit the generalizability of our results to other groups. This study showed a change in individuals’ levels of stress over the course of the study, but we were not able to evaluate longer-term outcomes. Future research should also investigate the impact of this type of intervention on persons from other age groups.

## 5. Conclusions

These results highlight the potential for digital health interventions to improve chronic disease outcomes beyond traditional education approaches ([45]). Given that individuals with low health literacy often struggle with complex medical information, providing content at an appropriate comprehension level may foster greater understanding and self-efficacy ([43]). The observed reduction in stress suggests that personalized content delivery may alleviate feelings of being overwhelmed and confused, leading to more confident health management behaviors. Future interventions should consider not only tailoring content to literacy levels but also incorporating additional personalization features, such as adaptive feedback, interactive elements, or real-time support, to further enhance engagement and outcomes. Future research should explore these additional mediating pathways as well as assess the long-term impact of digital tailoring on health outcomes. The advent of large language models may allow for highly sophisticated tailoring to multiple patient characteristics ([2]; [27]). Finally, further studies should examine the scalability of such interventions, particularly in underserved populations where chronic disease burden and health literacy challenges are most pronounced.

## Figures and Tables

**Figure 1 behavsci-15-00783-f001:**
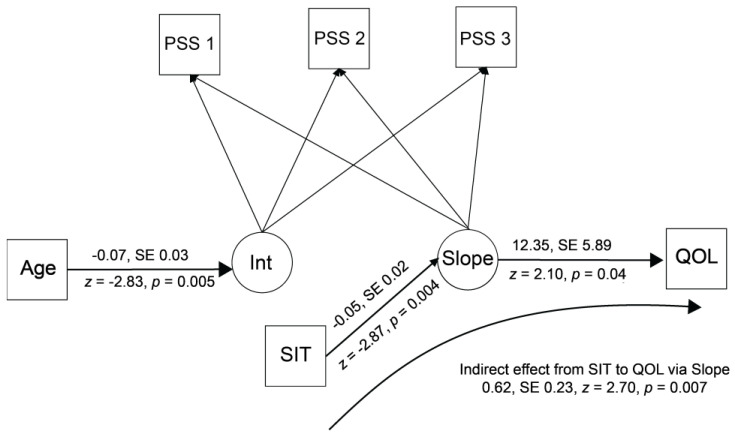
Final model. Note: PSS 1, 2, 3 = Perceived Stress Scale at baseline, first and second follow-ups; Age = Age in years; SIT = Success in Tailoring scale; QOL = SF-36 General Health subscale at study end; Int = Growth curve model intercept; Slope = Growth curve model slope.

**Table 1 behavsci-15-00783-t001:** Descriptive statistics for participants (n = 309).

Variable	Count
Men	144
Women	165
White	41
Black	268
	Mean (SD)
Age in Years	57.63 (8.41)
Education Years	11.86 (1.85)
Total Number of Health Conditions	6.47 (2.78)
HRQOL (SF General Health) ^1^	60.19 (19.66)
Perceived Stress Scale—Baseline	22.46 (3.81)
Perceived Stress Scale—Follow-Up 1	21.54 (3.76)
Perceived Stress Scale—Follow-Up 2	21.84 (3.27)

^1^ HRQOL = Health-related quality of life; SF General Health = Medical Outcomes Studies, Short Form-36 General Health subscale.

## Data Availability

The raw data supporting the conclusions of this article will be made available by the authors on request.

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
