# Peer review of "Change in Stress Among Individuals with Chronic Health Conditions and Low Health Literacy Using a Tailored Communication App Promoting Self-Management"

_behavsci, 2025, doi:10.3390/bs15060783_

Round 1
Reviewer 1 Report
Comments and Suggestions for Authors
Dear Authors, thanks for submitting this interesting study “Change in Stress Among Individuals with Chronic Health Con-2 ditions and Low Health Literacy Using a Tailored Communica-3 tion App Promoting Self-Management” . this study explores the impact of a tailored mobile health intervention on stress reduction and self-management among individuals with chronic diseases and low health literacy. Given the increasing role of digital health solutions in chronic disease management, the research question is timely and relevant. Also, this study aligns with the growing field of eHealth and digital self-management tools, particularly for individuals with low health literacy and chronic conditions. Additionallyt, this study have also a robust methodological approach as the use of latent growth curve modeling is appropriate for assessing longitudinal changes in stress.
However, several aspects could be refined. In particular, so, here below are detailed comments for improvement:
Introduction: this section provides an overview of health literacy and self-management but lacks a clear gap statement. Maybe the authors could explicitly state how this study extends previous research? Is the innovation in the tailored intervention itself? The focus on digital literacy and chronic disease? In the best of my knowledge, the introduction also lacks of an in-depth examination of references. There are few bibliographical references relating to the key points of the study. Especially in lines 29-41 many references related to the management of chronicity in terms of multimorbidity and polypharmacy are not really mentioned (..but were discussed). some examples are: doi: 10.1007/s10865-007-9147-y doi: 10.3390/ijerph18094422 doi: 10.1016/S2666-7568(24)00007-2 and so on…
Methods: The sample consists of 309 participants aged 40+ with low health literacy. How representative is this group? Were any exclusion criteria applied that could affect generalizability? Please explain more in depth this aspect in the methodplogy section.
Statistical Analysis: The authors mention “computational difficulties” with the initial model. Were any model assumptions violated? How was missing data handled? Please vould you more expalin this and, eventually, add it into the methodology section?
Discussion: The study focuses on short-term stress reduction. Did you also assess follow-up retention or sustained effects beyond the intervention period? Will you consider doing this in a future study? Perhaps it would be interesting to discuss it
Author Response
Reviewer 1 comments:
Dear Authors, thanks for submitting this interesting study “Change in Stress Among Individuals with Chronic Health Conditions and Low Health Literacy Using a Tailored Communication App Promoting Self-Management”. this study explores the impact of a tailored mobile health intervention on stress reduction and self-management among individuals with chronic diseases and low health literacy. Given the increasing role of digital health solutions in chronic disease management, the research question is timely and relevant. Also, this study aligns with the growing field of eHealth and digital self-management tools, particularly for individuals with low health literacy and chronic conditions. Additionally, this study has also a robust methodological approach as the use of latent growth curve modeling is appropriate for assessing longitudinal changes in stress.
However, several aspects could be refined. In particular, so, here below are detailed comments for improvement:
Introduction: this section provides an overview of health literacy and self-management but lacks a clear gap statement. Maybe the authors could explicitly state how this study extends previous research? Is the innovation in the tailored intervention itself? The focus on digital literacy and chronic disease?
Response: The introduction has been rewritten to clarify this issue.
In the best of my knowledge, the introduction also lacks of an in-depth examination of references. There are few bibliographical references relating to the key points of the study. Especially in lines 29-41 many references related to the management of chronicity in terms of multimorbidity and polypharmacy are not really mentioned (but were discussed). some examples are: doi: 10.1007/s10865-007-9147-y doi: 10.3390/ijerph18094422 doi: 10.1016/S2666-7568(24)00007-2 and so on…
Response: Thank you for your comments. We have edited the introduction to address both reviewer comments. We added additional information regarding multimorbidity and polypharmacy as suggested and its impact on adherence and health outcomes. Additional references were added.
Methods: The sample consists of 309 participants aged 40+ with low health literacy. How representative is this group?
Response: It’s not clear precisely what the reviewer means by “representative” in this context. We believe the sample is reasonably representative of a minoritized population with few educational and economic resources, and thus would not be representative of, for example, upper middle-class Whites. This is both a positive aspect of the study (providing information on the useful of tailored apps in persons 40 years of age and older with low health literacy) but limits the generalizability of the findings to other groups. This has been added to the review of study limitations in the discussion section of the MS.
Were any exclusion criteria applied that could affect generalizability? Please explain more in depth this aspect in the methodology section.
Response: No exclusion criteria were applied, except if the investigators at each site (both experienced clinicians) judged that the person was not able to provide informed consent for their participation. This has been added to the methods sections as suggested by the reviewer. This information has been added to the MS.
Statistical Analysis: The authors mention “computational difficulties” with the initial model. Were any model assumptions violated?
Response: Initial computations resulted in a nonpositive definite latent variable covariance matrix, with Mplus identifying the slope of the growth curve as the problematic variable. It had an impossible negative value. Consistent with others’ recommendations for handling this problem (Zitzmann et al., 2022), we assumed that the correct value was either zero or very small; we chose to fix the slope during initial estimation to a small positive value to allow calculation of other output variables (fixing it to zero would lead to division by zero errors in calculation of other variables). The latent variable covariance matrix was then positive definite and other calculations could continue.
Here is the error message from the software:
“WARNING: THE LATENT VARIABLE COVARIANCE MATRIX (PSI) IS NOT POSITIVE DEFINITE. THIS COULD INDICATE A NEGATIVE VARIANCE/RESIDUAL VARIANCE FOR A LATENT VARIABLE, A CORRELATION GREATER OR EQUAL TO ONE BETWEEN TWO LATENT VARIABLES, OR A LINEAR DEPENDENCY AMONG MORE THAN TWO LATENT VARIABLES. CHECK THE TECH4 OUTPUT FOR MORE INFORMATION. PROBLEM INVOLVING VARIABLE SPSS.”
(The variable “SPSS” was the name we assigned to the slope of the Perceived Stress Scale).
How was missing data handled? Please could you more explain this and, eventually, add it into the methodology section?
Response: Missing data were handled using full information maximum likelihood estimation. This strategy has repeatedly been shown to be robust (Enders & Bandalos, 2014). We have added this to the methods sections as suggested.
Discussion: The study focuses on short-term stress reduction. Did you also assess follow-up retention or sustained effects beyond the intervention period? Will you consider doing this in a future study? Perhaps it would be interesting to discuss it
We assessed change in stress from baseline to immediately after participants completed the intervention and then again three months later, as was originally planned in the grant, so we evaluated sustained effects over three months. We were unsuccessful in obtaining funding for a follow-up study. This point has also been added to the review of study limitations in the discussion.
Reviewer 2 Report
Comments and Suggestions for Authors
Please see the attached comments.

Author Response
Reviewer 2 (drawn from a PDF file):
The study addresses a critical topic, that is the need to develop tailored interventions that are accessible for individuals with chronic illness. A focus on self-management and integration of mobile technology is a strength. I also appreciated the authors’ attention to socioeconomic and minority status barriers as well as acknowledgement of a role for health literacy.
Response: Thank you for your comments. We have addressed suggestions and comments below to strengthen the manuscript. We hope it is clearer.
However, I found the paper at times difficult to read. Although the sections were all appropriate and logically placed, the information within the sections was not always organized well and often lacked sufficient explanation. These suggestions are intended to help improve the quality of the paper.
Response: Noted. We appreciate your comments and hope we have successfully addressed them.
- The Introduction starts off well highlighting barriers to adherence with complex regimen (low SES, distress, health literacy) and implications for low self-esteem, self-confidence and stress. This feeds back nicely into the constructs measured. The authors focus on a rationale for chronic disease management with a focus on self-management, which also resonates with the need to develop interventions that increase one’s effectiveness. But then things become more disjointed in the last paragraph of the Intro.
- Response: Thank you for your thoughtful review. We have edited the last paragraph to address a/b/c comments. Please see below for more detailed explanations. We added additional information in the beginning to address other reviewer comments. The final several sentences of the introduction are rewritten to clarify the purpose of the paper.
- a) Stress is introduced as impacting outcomes (1 sentence) but is not discussed (what type of stress, general life stressors or illness specific) and with no cited literature to support the association with any of the other constructs examined.
Response: Additional information regarding different types of stressors have been added and how that relates to allostatic load. Related references are now provided. We hope that this section is now clearer and better documented.
- b) You quickly move to relay that there’s a need to develop interventions “at appropriate level”, but there’s not elaboration on what specifically this means in context of this study. This is also not tied into other constructs being examined (health literacy, self-efficacy, stress).
- Response: Additional information referencing review and meta-analysis regarding how information should be tailored to the appropriate health literacy level but also targeted to a patient’s specific needs, and culturally adapted to improve use. Additionally, a sentence was added regarding the use of digital tools to develop content using these constructs to manage stress and promote disease self-management (again with appropriate references).
- c) The paragraph ends with Aims but they’re hard to follow (e.g., refer to outcomes but don’t list them, no mention of health literacy; examine “relationships” but directionality not clear). Please conclude with a stronger rationale for this secondary analysis and outline the aims specifically (e.g., questions to answer, hypotheses).
As noted above, we have completely rewritten the problematic paragraph to clarify these issues.
- There are 3 different versions of the app mentioned aligned with different reading levels. Please clarify how this was determined and how you matched participants to levels. If it’s randomized, then you’re not really “tailoring” or individualizing the intervention.
- How did you decide on the chosen age of 40 years and older. Seems random (why not adults 18 and up) without further justification.
- Participants engaged in 3 sessions over 2-3 weeks. Please clarify how the content was split across the 3 sessions (what was the focus of each).
- Did you define “chronic health condition” for participants since this was the primary inclusion criteria? What was the definition since this can be interpreted differently (different severity, different duration, impact on functioning, etc.)?
- It is stated that the intervention/app content was tailored. Since this is a key element of your study, please explain how things were individualized (perhaps give some examples).
- All of these issues are covered in detail in the primary outcome publication as well as a separate publication on how the intervention was developed. These are available as open access publications and we have uploaded them as additional material for review. Our aim is to avoid duplicate publication of the same material, especially since the other papers provide much greater detail.
- f. The “outcomes” section can be organized better. It was hard to follow to fully appreciate what outcomes you were examining and very little information was provided about the measures.
We regret not having provided sufficient information. We note that the primary measures presented in the analyses are the Perceived Stress Scale and the SF-36 and the scale we created to measure the relevance of information to the participant. Since the first two are widely used, in the original manuscript we only discussed the scale we refer to as the Success in Tailoring (SIT) scale. We have inserted additional information about the other two scales
- How was adherence measured? Is self-report noted as yes/no adherent or degree of adherence? Adherence with what aspect of their regimen?
In response to this reviewers suggestion that we remove discussion of measures not directly linked to this study, we have removed mention of adherence.
- Say more about the “relevance measure” (what’s the content focus, how many items, yes/no relevant vs. degree of relevance, relevant to what, etc.).
- As noted in the MS, the measure was based on a review of the concept of relevance.
We thus aimed to assess the extent to which information presented was perceived as related to the person, was appropriate to their individual situation, was usable, and whether they intended to act on it. The measure comprised eight items rated on a five-point Likert-type scale. Additional information has been added to the MS
- There’s reference to stress levels here but no information is provided (what type of stress, stress due to daily stressors vs. medical regimen)
- Reference to secondary outcomes was vague; if not going to discuss them consider removing that paragraph
- Results
- I’m curious if you examined demographic differences in relation to the other variables? Gender? Age? Racial differences in the outcomes? The high % of Black participants relative to White is notable. If we’re going to “tailor” interventions and attend to SES related barriers, it’s important to examine group differences.
- The Table includes many measures that are not utilized in this particular study which is just focused on secondary analyses; either explain how they are utilized or remove them.
- As noted above, in response to this reviewer’s comments, we have removed the material from the table as well as the text.
- The Discussion section was very strong, explaining the results and highlighting the clinical implications. This was by far the more organized, clear, elaborated section of the paper.
We thank the reviewer for his or her positive comments.
Reviewer 3 Report
Comments and Suggestions for Authors
I would like to thank the editors for the opportunity to review this manuscript.
This study is well-conducted and specifically examines the role of tailored digital health interventions in reducing stress and improving quality of life, particularly among individuals with low health literacy. Additionally, the study developed applications and surveys to investigate the relationship between perceived personalization of app-delivered information, stress reduction, and self-management. The key findings not only reveal the nuanced relationship between these factors but also provide effective strategies for improving disease management.
Major comments:
1.The introduction provides a well-rounded and critical review of various aspects of chronic disease self-management (CDSM). However, the justification for using tailored digital health interventions needs further discussion, particularly regarding their specific advantages over traditional approaches. It would be beneficial to clarify which aspects such as patient engagement, adherence, or accessibility make digital interventions more effective. Additionally, while meta-analyses provide valuable insights, incorporating findings from empirical studies, such as randomized controlled trials or longitudinal research, would strengthen the argument.
- 2.1. App Development. The development of the mobile application has been thoroughly described. However, incorporating screenshots of the application would enhance readers' understanding of its functionality and user interface. Additionally, further clarification of the user involvement process is recommended. Specifically, it would be beneficial to provide details on the number of volunteers or experts who participated in the testing, as well as specify the dates when the usability testing was conducted.
- 2.3. Participant Recruitment and Screening. Clarifying the number of participants initially recruited and those excluded due to higher health literacy levels is recommended. Providing this information would enhance the study’s methodological rigor and improve transparency in participant selection and data interpretation.
- 2.4 outcome measure. The authors utilized measurement instruments developed strictly from prior studies. However, the study reports only reliability testing without assessing validity (e.g., convergent or discriminant validity). The absence of validity testing raises concerns about whether the measures accurately capture the intended constructs. If further validity testing is not required, the authors should justify this decision and provide evidence supporting the appropriateness of the adopted measures.
- Ethics Approval. The date of ethics approval is missing. Including this information in this section would improve the clarity and completeness of the study's ethical considerations.
- Discussion. The study's participants are limited to individuals aged 40 years and older with low health literacy. Shouldn't other age groups with low health literacy also be considered for future investigation? Moreover, the study sample exhibits an unequal distribution of ethnicities, with a higher representation of Black participants compared to White participants. Acknowledging these limitations is recommended.
Minor comments:
- The abstract should include the total number of participants, the age range, and the statistical package used in the study.
- The manuscript should present separate discussion and conclusion sections rather than combining them, as merging both results in an overly dense discussion that is difficult to read.
With appropriate refinements to address the major concerns outlined above, this manuscript could make a meaningful contribution to the relevant field of study and health interventions.
Author Response
Reviewer 3 comments:
I would like to thank the editors for the opportunity to review this manuscript.
This study is well-conducted and specifically examines the role of tailored digital health interventions in reducing stress and improving quality of life, particularly among individuals with low health literacy. Additionally, the study developed applications and surveys to investigate the relationship between perceived personalization of app-delivered information, stress reduction, and self-management. The key findings not only reveal the nuanced relationship between these factors but also provide effective strategies for improving disease management.
Major comments:
1.The introduction provides a well-rounded and critical review of various aspects of chronic disease self-management (CDSM). However, the justification for using tailored digital health interventions needs further discussion, particularly regarding their specific advantages over traditional approaches.
This material has been added to the introduction.
It would be beneficial to clarify which aspects such as patient engagement, adherence, or accessibility make digital interventions more effective. Additionally, while meta-analyses provide valuable insights, incorporating findings from empirical studies, such as randomized controlled trials or longitudinal research, would strengthen the argument.
As meta-analyses are typically based on clinical trials it’s not clear to us how to respond to this comment. In the introduction, however, in response to other reviewers’ comments, citations to several clinical trials have been added.
- 2.1. App Development. The development of the mobile application has been thoroughly described. However, incorporating screenshots of the application would enhance readers' understanding of its functionality and user interface. Additionally, further clarification of the user involvement process is recommended. Specifically, it would be beneficial to provide details on the number of volunteers or experts who participated in the testing, as well as specify the dates when the usability testing was conducted.
All of this information is provided in other publications. In order to avoid duplicate publication of previously published information, we now refer the reader to the original publications and we have provided copies of the original publications as additional material for review.
- 2.3. Participant Recruitment and Screening. Clarifying the number of participants initially recruited and those excluded due to higher health literacy levels is recommended. Providing this information would enhance the study’s methodological rigor and improve transparency in participant selection and data interpretation.
This information is provided in the publication of the main study outcomes, including a CONSORT diagram detailing precisely these numbers. This open access paper has been uploaded as additional material for review for the convenience of the reviewers.
- 2.4 outcome measure. The authors utilized measurement instruments developed strictly from prior studies. However, the study reports only reliability testing without assessing validity (e.g., convergent or discriminant validity). The absence of validity testing raises concerns about whether the measures accurately capture the intended constructs. If further validity testing is not required, the authors should justify this decision and provide evidence supporting the appropriateness of the adopted measures.
The Perceived Stress Scale and the SF-36 have been widely used and are well validated, and they were not developed by us. The only measure for which we report only reliability information is the one developed for the purpose of this study, to which we refer as “Success in Tailoring Scale.” In the timeline of developing the app, it was not possible to do a full development of the scale while creating the intervention and completing usability testing. As for convergent validity, the reason we created the scale was because we could not find any other scale for this purpose, so that it was not possible to measure the extent to which this measure assessed the same construct as another. In the absence of convergent validity, it is not clear what purpose demonstrating that the measure is unrelated to other measures would serve.
While creating the scale, we focused on finding a conceptual basis for the items and ensuring that they had, at least, face validity. We have added additional information to the section discussing the material, including several example items, so that the reader can judge for him or herself whether the measure is valid for the purpose usee here.
- Ethics Approval. The date of ethics approval is missing. Including this information in this section would improve the clarity and completeness of the study's ethical considerations.
Dates were included in the ethics approval statement at the end of the paper as specified by the journal instructions. We have copied this statement to the section of the MS the reviewer refers to.
- Discussion. The study's participants are limited to individuals aged 40 years and older with low health literacy. Shouldn't other age groups with low health literacy also be considered for future investigation? Moreover, the study sample exhibits an unequal distribution of ethnicities, with a higher representation of Black participants compared to White participants. Acknowledging these limitations is recommended.
We have added the suggested comments to the discussion section.
Minor comments:
- The abstract should include the total number of participants, the age range, and the statistical package used in the study.
We have added this information to the abstract.
- The manuscript should present separate discussion and conclusion sections rather than combining them, as merging both results in an overly dense discussion that is difficult to read.
We have added a separate heading for the conclusion.
With appropriate refinements to address the major concerns outlined above, this manuscript could make a meaningful contribution to the relevant field of study and health interventions.
Round 2
Reviewer 2 Report
Comments and Suggestions for Authors
Thank you for the opportunity to review the revised manuscript. The authors made a number of changes that improved the clarity and readability. The introduction has more depth, particularly with regard to stress. The aims are more clear with a stronger rationale.
The newly revised paragraph at the end of the Introduction is extremely long -- ideas about the impact of stress, available interventions, and study aims-- are all merged together. To increase readability consider separating them into smaller paragraphs.
There is an error at the end of the Aims paragraph: "Individually tailored information had an "income" on study outcomes". I thin you mean "impact"?
There is a sentence at the end of the Participants section that seems to be misplaced: "This has been added to the methods section as suggested by the reviewer" does not belong anywhere in the paper, just in the author response letter.
The comment about examination of demographic differences was not addressed. This is important in order to understand effectiveness of interventions and generalizability of the findings.
Some of the previous questions in the Methods section were not address. Regarding the authors' response that the information can be found in the original publication, I do not believe this is appropriate particularly as it relates to the intervention, which is the focus of this study. I will leave this decision up to the Editor.
Author Response
Revision – Response to Reviewer
Thank you for the opportunity to review the revised manuscript. The authors made a number of changes that improved the clarity and readability. The introduction has more depth, particularly with regard to stress. The aims are more clear with a stronger rationale.
The newly revised paragraph at the end of the Introduction is extremely long -- ideas about the impact of stress, available interventions, and study aims-- are all merged together. To increase readability consider separating them into smaller paragraphs.
We have revised this paragraph as suggested (line 91 to 127).
There is an error at the end of the Aims paragraph: "Individually tailored information had an "income" on study outcomes". I thin you mean "impact"?
We regret the oversight and have correct this word (line 123).
There is a sentence at the end of the Participants section that seems to be misplaced: "This has been added to the methods section as suggested by the reviewer" does not belong anywhere in the paper, just in the author response letter.
The erroneous sentence has been removed from the MS (lines 179-180).
The comment about examination of demographic differences was not addressed. This is important in order to understand effectiveness of interventions and generalizability of the findings.
In the description of model building, we note that race, gender and age were included in preliminary models but only age was significantly related to either baseline level of stress and none of them were related to change in stress. We have now added a sentence noting that we did not observe differences related to gender or race in main study outcomes. Age was related to patient activation and education was related to health-related quality of life. We refer to the open access paper with full study results for interested readers (lines 255-257).
Some of the previous questions in the Methods section were not address. Regarding the authors' response that the information can be found in the original publication, I do not believe this is appropriate particularly as it relates to the intervention, which is the focus of this study. I will leave this decision up to the Editor.
The reviewer’s previous comments on methodology:
- There are 3 different versions of the app mentioned aligned with different reading levels. Please clarify how this was determined and how you matched participants to levels. If it’s randomized, then you’re not really “tailoring” or individualizing the intervention.
We have added additional clarification on how information was tailored in a paragraph at the end of the section on app development (lines 143-157).
- How did you decide on the chosen age of 40 years and older. Seems random (why not adults 18 and up) without further justification.
Since the focus of the intervention was on individuals with chronic health conditions, only individuals 40 years of age and older were included because of the low prevalence of chronic health conditions in younger individuals (lines 172-175).
- Participants engaged in 3 sessions over 2-3 weeks. Please clarify how the content was split across the 3 sessions (what was the focus of each).
The content of the intervention was distributed over three sessions. The first day included an introduction to the app, its purpose, and its functioning and modules on adherence and stress. The second day focused on sleep, mood, pain, and memory, while the third focused on fatigue, shortness of breath, and anger. (lines 154-157).
- Did you define “chronic health condition” for participants since this was the primary inclusion criteria? What was the definition since this can be interpreted differently (different severity, different duration, impact on functioning, etc.)?
Chronic conditions were defined as a diagnosis and current treatment for one of the items in the Functional Comorbidity Index. This was done because the Index was defined as including conditions related to physical function. It was expanded to include a number of conditions requiring treatment that are common in US Medicare statistics, such as hypertension and dyslipidemia (lines 180-185).
- It is stated that the intervention/app content was tailored. Since this is a key element of your study, please explain how things were individualized (perhaps give some examples).
We have added a paragraph at the end of the section on app development that explains the tailoring process and provides examples (lines 143-157).